# Sex Differences in Cardiovascular Disease Mortality in Brazil between 1996 and 2019

**DOI:** 10.3390/ijerph191912827

**Published:** 2022-10-07

**Authors:** Antonio de Padua Mansur, Desidério Favarato, Célia Maria Cassaro Strunz, Solange Desirée Avakian, Antonio Carlos Pereira-Barretto, Edimar Alcides Bocchi, Luiz Antonio Machado César

**Affiliations:** 1Serviço de Prevencao, Cardiopatia na Mulher e Reabilitação Cardiovascular, Instituto do Coracao (InCor), Hospital das Clinicas HCFMUSP, Faculdade de Medicina, Universidade de Sao Paulo, Sao Paulo 05403-900, SP, Brazil; 2Unidade Clinica de Aterosclerose, Instituto do Coracao (InCor), Hospital das Clinicas HCFMUSP, Faculdade de Medicina, Universidade de Sao Paulo, Sao Paulo 05403-900, SP, Brazil; 3Laboratorio de Analises Clinicas, Instituto do Coracao (InCor), Hospital das Clinicas HCFMUSP, Faculdade de Medicina, Universidade de Sao Paulo, Sao Paulo 05403-900, SP, Brazil; 4Unidade Clínica de Valvopatias, Instituto do Coracao (InCor), Hospital das Clinicas HCFMUSP, Faculdade de Medicina, Universidade de Sao Paulo, Sao Paulo 05403-900, SP, Brazil; 5Unidade Clinica de Insuficiencia Cardiaca, Instituto do Coracao (InCor), Hospital das Clinicas HCFMUSP, Faculdade de Medicina, Universidade de Sao Paulo, Sao Paulo 05403-900, SP, Brazil; 6Unidade Clinica de Coronariopatias Cronicas, Instituto do Coracao (InCor), Hospital das Clinicas HCFMUSP, Faculdade de Medicina, Universidade de Sao Paulo, Sao Paulo 05403-900, SP, Brazil

**Keywords:** all causes of death, cardiovascular diseases, ischemic heart diseases, stroke, mortality, women, men

## Abstract

Background: cardiovascular diseases (CVD) are Brazil’s leading causes of death in women and men. This study analyzed age-adjusted death rate (DRaj) trends from all causes of death (ACD), CVD, ischemic heart disease (IHD), and stroke in women and men aged 35 to 74 years from 1996 to 2019. Methods: We analyzed DRaj trends for all causes of death (ACD), CVD, IHD, and stroke. Data were from the Ministry of Health mortality database. Joinpoint Regression Program™ performed trend analysis and adjustments in death rates. Average annual percentage change (AAPC) determined the intensity of changes. Results: In women, DRaj reduced for ACD (AAPC = −1.6%); CVD (AAPC = −2.6%); IHD (AAPC = −1.9%); and stroke (AAPC = −4.6%) (*p* < 0.001 for all). In men, ACD reduced from 1996 to 2004 (AAPC = −0.9%; *p* < 0.001), from 2012 to 2019 (AAPC = −1.9%; *p* < 0.001), and unchanged from 2004 to 2012; CVD (AAPC = −2.1%); IHD (AAPC = −1.5%); stroke (AAPC = −4.9%) (*p* < 0.001 for all) reduced from 1996 to 2019. From 1996 to 2019, the male/female ratio for ACD remained unchanged. CVD increased from 1.58 to 1.83, IHD from 1.99 to 2.30, and stroke from 1.52 to 1.83. Conclusion: ACD, CVD, IHD, and stroke were reduced more significantly in women, and the ratio of CVD, IHD, and CVD in men and women increased more in men. Future studies will be needed to determine the main factors responsible for a better outcome in women.

## 1. Introduction

Cardiovascular diseases (CVD) are currently the leading cause of death worldwide and in the Brazilian population [1]. The age-standardized death rate from CVD, IHD, and stroke decreased by 10.3%, 9.7%, and 13.6% worldwide, respectively, from 2007 to 2017. The most significant decrease in percentage change occurred in the highest-income regions, probably due to improvements in prevention policies and health care accessibility. IHD and stroke accounted for almost 85% of all cardiovascular deaths, and the death rate was higher in men than in women, except in the age group over 80 years [2]. In South America, the leading causes of death from noncommunicable diseases were attributed to CVD (31.1%), cancer (30.6%), and respiratory diseases (8.6%), with a higher CVD death rate in men than in women [3]. In Brazil, since 1986, there has been a gradual reduction in the death rate from these diseases [4]. The decline of stroke was gradual and persistent. However, IHD in women and men had a period of stagnation in the death rate between 2007 and 2012 [5], and the same deceleration trend was observed for IHD mortality in the USA [6]. After this period, the IHD death rate in Brazil returned to its downward trend. The decline in IHD and stroke death rate is not well known yet. However, the main hypotheses are the improved care of primary risk factors with access to free or subsidized drugs for hypertension, diabetes, and dyslipidemia, and better access to public hospitals with improved guideline-based care for acute cardiovascular events [7]. Also, some Brazilian government policies significantly reduced tobacco consumption [8] and favored the increasing adoption of a healthy diet and physical activity by the population. Nevertheless, we are observing an increase in the prevalence of obesity and diabetes [9]. In 2016, 17% of Brazilians were obese [9]. The changes in the prevalence of these risk factors may impact future cardiovascular morbidity and mortality. It is well known that atherosclerosis, the primary pathological process responsible for cardiovascular disease, manifests differently in women and men. Women have up to ten years of advantage over men in the incidence of CVD [10]. The estrogen and its derivatives present in the premenopausal period probably mediate the lower incidence of CVD in women. The endogenous estrogen promotes vascular health through beneficial genomic and nongenomic mechanisms. A recent study on the Brazilian population aged 35 to 74 years between 1996 and 2017 showed that the age-adjusted death rate (DRaj) from CVD corresponded, on average, to 31% of deaths from all causes, but the death rate from CVD, IHD, and stroke was higher in men than in women [11]. Between 1996 and 2017, we observed a 38% reduction in CVD mortality in the Brazilian population. The reduction percentage was more pronounced in women than in men, respectively, at 41% and 35%, similar to the results observed in the Global Burden of Disease Study-2017 [12]. In Brazil, women had the lowest death rate and the highest percentage of reduction in CVD, IHD, and stroke. The same result was observed, for example, in the USA [6,13]. CVD mortality in women has decreased globally in most developed countries in the last two decades. However, mortality remained unchanged in countries with a low sociodemographic index [14]. Similarly, we do not have more recent data on trends in death rates and mortality in women and men in our population. The objective of this study was to update and analyze the trends in DRaj for the 35–74 age group and the average annual percentage change (AAPC) for all causes of death, CVD, IHD, and stroke, in women and men in the Brazilian population between 1996 and 2019. These sex-related analyses may improve public policies to reduce the death rate from CVD even more in Brazil.

## 2. Materials and Methods

We analyzed the trends of all causes of death (ACD), CVD, IHD, and stroke in men and women in Brazil from 1996 to 2019. We analyzed the crude death rate per 100,000 population for each five-year age group between 35 and 74 years. We calculated the DRaj for the 35–74 age group per 100,000 population for the study period (1996–2019) by the direct method using the World Health Organization (WHO) 2000 world standard population. We obtained the mortality data from the Vital Statistics of Health Information on the DATASUS page of the Ministry of Health’s online mortality database [15]. The causes of death were classified by the 10th revision of the Classification International on Disease. We grouped the CVD under codes I00 to I99, IHD under codes I20 to I25, and stroke under codes I60 to I69. The study does not require analysis by the Ethics Committee, as the mortality data were obtained from a public website and without identifying the individuals.

### Statistical Analysis

We used the statistical program Joinpoint Regression Program version 4.7.0.0 from the National Cancer Institute, Division of Cancer Control and Population Sciences (Bethesda, Maryland, USA), to analyze trends in the age-adjusted death rate from 35 to 74 years [16]. The junction point analysis or segmented regression modeling was used to identify the years (independent variable) in which there were significant changes in the mortality rate (dependent variable) from 1996 to 2019. This method identifies joinpoints connecting distinct line segments, thus characterizing changes over time. The AAPC determined the intensity of the changes in the death rate. Comparisons of the linear regression line slopes (CLRLS) were analyzed in Microsoft Excel 2010 using the t statistic and two-tailed t distribution [17]. The necessary assumptions of linear regression analysis were verified, such as linearity, homoscedasticity, error independence, non-multicollinearity, and low exogeneity. Statistical significance was for *p*-values < 0.05.

## 3. Results

The DRaj, the AAPC, and the junction points (Jp) by ACD, CVD, IHD, and stroke, in men and women are shown in Table 1 and Figure 1.

The DRaj for ACD, CVD, IHD, and stroke were higher in men. ACD decreased from 1996 to 2019 (0 Jp; AAPC = −1.6%; *p* < 0.001). ACD in men reduced from 1996 to 2004 (AAPC = −0.9%; *p* < 0.001), remained unchanged from 2004 to 2012, and reduced from 2012 to 2019 (3 Jp; AAPC = −1.9%; *p* < 0.001). In women, ACD decreased from 1996 to 2019 (0 Jp; AAPC = −1.6%; *p* < 0.001). The DRaj for total CVD, CVD in men, and CVD in women reduced and the AAPC were, respectively, −2.4% (*p* < 0.001), −2.1% (*p* < 0.001) and −2.6% (*p* < 0.001). IHD and stroke declined in women and men from 1996 to 2019, but stroke reduction in men was more significant from 1996 to 2001 than from 2001 to 2019 (*p* < 0.001).

The slopes of the regression line for the age group 35 to 74 years, in women and men, for ACD, CVD, IHD and stroke showed a significant reduction in the death rate from these diseases from 1996 to 2019 in both sexes (Figure 2).

The CLRLS between women and men showed a more significant reduction in the death rate from ACD, CVD, IHD, and stroke in men (*p* < 0.0001 for all).

The percentual proportion of total CVD from ACD reduced by 15% from 1996 to 2019, respectively, 33% to 28%; in men, it was 32% to 29%, with a reduction of 9%, and in women, it was 35% to 28%, with a decrease of 22% (CLRLS: men vs. women; *p* = 0.127). In men, the proportion of IHD from CVD increased by 14% between 1996 and 2019, respectively, 35% and 41%, and in women, it was 28% and 33%, with an increase of 15% (CLRLS: men vs. women; *p* = 0.146). In men, the proportion of stroke from total CVD was 23% and 14% between 1996 and 2019, with a decrease of 37%. It was 24% and 15% in women, with a reduction of 39% (CLRLS: men vs. women: *p* = 0.589) (Table 2).

The age-adjusted male/female mortality ratio for CVD, IHD, and stroke increased from 1.59 to 1.84, 2.00 to 2.30, and 1.53 to 1.83 but remained unchanged for ACD from 1996 to 2019, respectively (Figure 3).

Crude death rates per 100,000 inhabitants for five-year age groups between 35 and 74 years, the percentage change in rates (1996–2019), and AAPC for ACD, CVD, IHD, and stroke, in the whole population, in men and women, are shown in Table 3 and Table 4.

There was a significant reduction in the crude death rate for ACD, CVD, IHD, and stroke from 1996 to 2019 (*p* < 0.001) in the total population, in men and women. The reduction in the crude death rate was more significant for CVD than for ACD (*p* < 0.001) and for stroke compared to IHD (*p* < 0.001) for the total population, in women and men. However, the detailed Jp analysis of crude mortality rates for CVD, IHD, and stroke in the whole population, in women and men, remained unchanged for some age groups in specific periods. For example, the crude death rate for the entire population aged 40–44, 50–54, and 65–69 years remained unchanged for CVD from 2005 to 2008, 2014 to 2019, and 2004 to 2008, respectively. In Appendix A, these data for the whole population, women, and men, are shown in Appendix A.

## 4. Discussion

This study showed a significant reduction in the age-adjusted death rate for ACD, CVD, IHD, and stroke in women and men. The percentage reduction in CVD, IHD, and stroke was more significant in women than men. These results were similar to those observed globally [12]. CVD decreased from 1990 onwards for all regions of the world, except for Sub-Saharan Africa and Oceania, where no significant change occurred, although in Central Asia, conversely, there was an increase of CVD mortality by almost 10% [18]. Globally, CVD in women in 2019 was estimated at 204 deaths per 100,000 population, with a 35.1% reduction since 1990 [19]. Our data showed that the death rate was 149 deaths per 100,000 population, with a decrease of 45% since 1996. The USA had the same decline in the CVD death rate. There was a 15% reduction in the CVD death rate from 2007 to 2017. IHD decreased, respectively, in men and women by 18% and 22%, and stroke by 17% in both sexes from 2006 to 2016. In 2016, IHD and stroke deaths were higher in men than women. [20]. Several European countries observed the same reduction in CVD mortality [21]. In the European Society of Cardiology member countries, CVD corresponded to 47% and 39% of all deaths in women and men, respectively. IHDs were responsible for 38% of CVD deaths in women and 44% in men. Stroke was the second most common cause of CVD deaths, accounting for 26% of all CVD deaths in women and 21% in men. We observed similar percentages in our study for IHD, accounting for 33% of CVD deaths in women and 41% in men in 2019. However, CVD deaths were lower in the Brazilian population than in the European population, probably because of the high number of older people in the European population. Likewise, the age group analyzed in our study, between 35 and 74 years, excluded the older population aged more than 75 years.

CVD, IHD, and stroke proportions were consistently higher in men than women. We observed an increasing proportion of death from IHD compared to stroke and an increase in the proportion of CVD, IHD, and stroke in men compared to women. We also observed increasing participation in the death rate from IHD and stroke in men. The IHD was always at least twice as high in men for the age group from 35 to 74 years. In this age group, at least three hypotheses justify men’s increase in IHD and stroke. The first hypothesis is the lower prevalence of primary risk factors for CVD in women aged 35 to 74 years. The hypertension prevalence in women in this age group was lower than in men, despite the global rise in hypertension. Undiagnosed hypertension and the percentage of treated hypertension were also higher in men than women. In Latin America and the Caribbean, 35% of women and 23% of men with hypertension had their blood pressure under control in 2019 [22]. Likewise, risk factors, dyslipidemia, and smoking were less prevalent in women [23,24,25]. Conversely, diabetes was more prevalent in women, and its incidence has increased in recent years. This fact may impact future mortality from chronic degenerative diseases [26]. Second, the more significant reduction of deaths due to ill-defined causes, which are more prevalent in men, may also justify the increase in IHD and stroke [27]. Third, compared to men, women have natural protection of 7 to 10 years in the incidence of CVD, probably related to estrogen presence in the premenopausal period [10]. Therefore, CVD occurs later in women, and the adjustment of CVD death rates for the age group 35 to 74 years will include a group of women with a lower incidence of IHD and stroke compared to the age groups over 75 years.

### Study Limitations

Despite the improvement in mortality data in Brazil, the main limitation of this study remains the low quality of death certificates, exemplified by errors related to diagnosis and accuracy, deaths associated with unknown causes, and errors in data entry. The number of death certificates with diagnoses based on symptoms, signs, and abnormal clinical and laboratory findings indirectly indicates data quality limitations. Despite the progressive improvement, such certificates are still significant in Brazil’s northeast, north, and center-west regions. Likewise, validation studies of mortality data are unavailable in most states and cities in the country.

## 5. Conclusions

We observed a significant reduction in the death rate in Brazil due to CVD, IHD, and stroke for women and men. There was also a growing prevalence of IHD and stroke in men more than women. These findings are essential for public health policies, as they showed that despite the downward trend in CVD in women and men, these diseases continue to be one of the leading causes of death in the Brazilian population, with more intensity in men than women. Therefore, governmental policies should prioritize CVD primary and secondary prevention for women and men to decrease the CVD death rate by facilitating the population’s access to health services for adequate diagnosis and treatment. Likewise, we should encourage the population’s adherence to a healthy life and the medications recommended by current guidelines for preventing and treating CVD.

## Figures and Tables

**Figure 1 ijerph-19-12827-f001:**
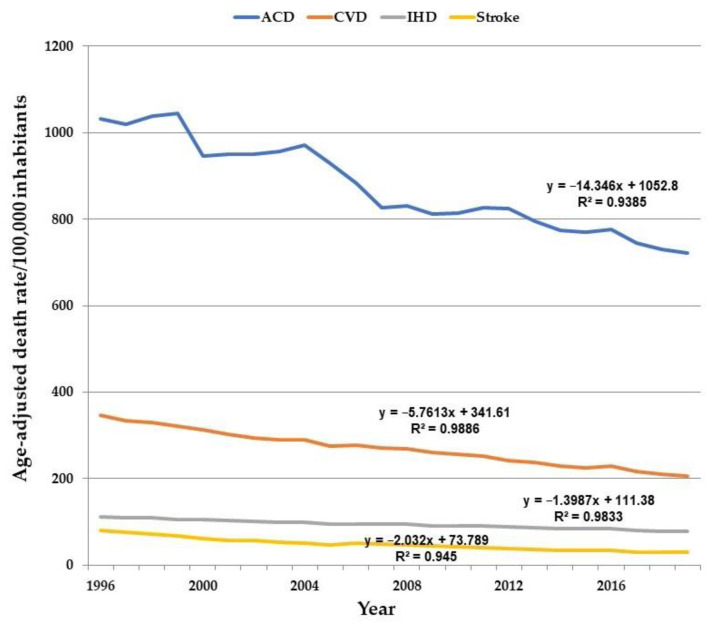
Adjusted death rates per 100,000 population, and regression data, for the age group 35 to 74 years for all causes of death (ACD), cardiovascular disease (CVD), ischemic heart disease (IHD), and stroke for the entire population from 1996 to 2019.

**Figure 2 ijerph-19-12827-f002:**
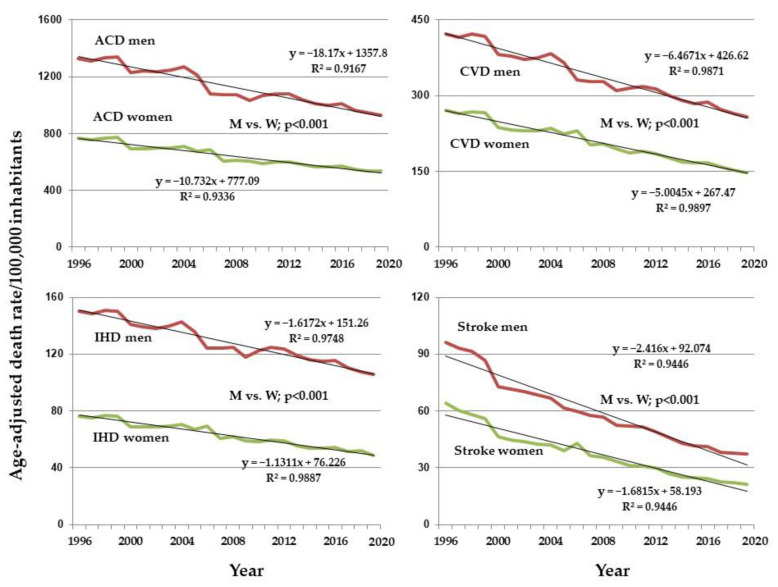
Comparisons of adjusted death rates regression lines per 100,000 population and regression data for all causes of death (ACD), cardiovascular disease (CVD), ischemic heart disease (IHD), and stroke in women and men from 1996 to 2019.

**Figure 3 ijerph-19-12827-f003:**
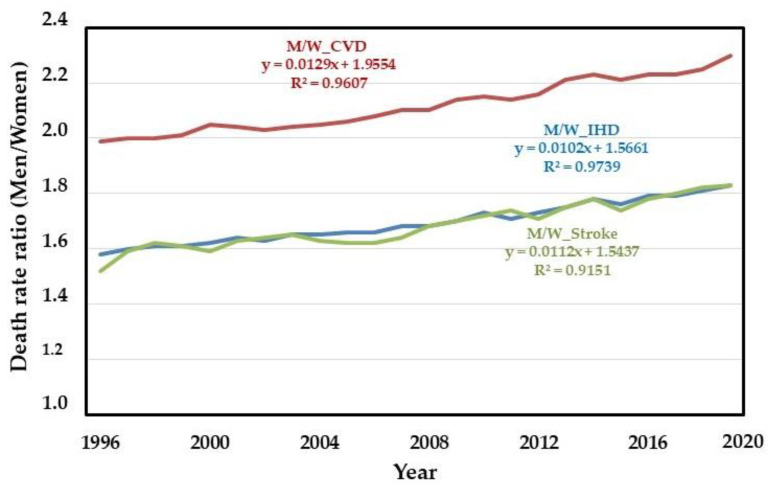
Regression lines of the ratio of men and women death rates per 100,000 population, and regression data, for cardiovascular disease (CVD), ischemic heart disease (IHD), and stroke in women (W) and men (M) from 1996 to 2019.

**Table 1 ijerph-19-12827-t001:** Adjusted death rates (DRaj) per 100,000 population for the age group 35 to 74 years, the number of Jp, and the mean annual percentage change (AAPC) for all causes of death (ACD), cardiovascular disease (CVD), ischemic heart disease (IHD) and stroke in men and women from 1996 to 2019.

Cause of Death	Year	DRaj	Year	DRaj	AAPC	99%CL	
ACD—0 Jp	1996	1032.2	2019	721.79	−1.6 *	−1.8	−1.5
ACD Men—3 Jp	1996	1327.4	2004	1268.2	−0.9 *	−1.6	−0.1
ACD Men—3 Jp	2004	1268.2	2007	1071.8	−5.1	−11	1.1
ACD Men—3 Jp	2007	1071.8	2012	1077	0.2	−1.7	2.1
ACD Men—3 Jp	2012	1077	2019	930.72	−1.9 *	−2.7	−1.2
ACD Women—0 Jp	1996	764.76	2019	721.79	−1.6 *	−1.8	−1.5
CVD—0 Jp	1996	342.85	2019	199.47	−2.4 *	−2.5	−2.2
CVD Men—0 Jp	1996	421.96	2019	258.03	−2.1 *	−2.3	−2
CVD Women—0 Jp	1996	270.84	2019	148.23	−2.6 *	−2.8	−2.4
IHD—0 Jp	1996	111.46	2019	75.15	−1.7 *	−1.9	−1.5
IHD Men—0 Jp	1996	150.23	2019	105.6	−1.5 *	−1.7	−1.3
IHD Women—0 Jp	1996	76.25	2019	48.29	−1.9 *	−2.1	−1.7
Stroke—0 Jp	1996	79.19	2019	28.62	−4.3 *	−4.6	−4.1
Stroke Men—1 Jp	1996	96.02	2001	71.57	−5.9 *	−7.6	−4.1
Stroke Men—1 Jp	2001	71.57	2019	37.1	−3.8 *	−4.1	−3.5
Stroke Women—0 Jp	1996	64.31	2019	21.39	−4.6 *	−4.9	−4.4

* *p* < 0.01. Jp: joinpoint.

**Table 2 ijerph-19-12827-t002:** Adjusted death rate (DRaj) per 100,000 population for the age group 35 to 74 years of the general population, in men (M) and women (W), from all causes of death (ACD), cardiovascular diseases (CVD), ischemic heart disease (IHD), and stroke per 100,000 inhabitants in Brazil from 1996 to 2019.

Year	ACD	ACD Men	ACD Women	M/W ACD	CVD	CVD Men	CVD Women	M/W CVD	IHD	IHD Men	IHD Women	M/W IHD	Stroke	Stroke Men	Stroke Women	M/W Stroke
1996	1032.24	1327.36	764.76	1.73	345.65	430.63	271.29	1.58	112.32	152.87	76.53	1.99	80.4	98.8	64.58	1.52
1997	1018.8	1310.72	754.19	1.73	332.98	417.15	259.53	1.6	108.66	148.25	73.8	2.00	75.53	94.43	59.36	1.59
1998	1038.12	1335.37	768.78	1.73	329.61	414.23	255.96	1.61	108.31	147.65	73.72	2.00	71.74	90.55	55.65	1.62
1999	1044.04	1341.6	774.38	1.73	320.31	402.7	248.95	1.61	105.67	144.5	71.62	2.01	66.97	84.37	52.23	1.61
2000	947.05	1230.61	694.99	1.77	312.32	393.65	242.08	1.62	105.11	144.96	70.37	2.05	60.62	75.93	47.66	1.59
2001	950.56	1238.86	694.44	1.78	302.6	383.43	233	1.64	102.42	141.02	68.88	2.04	57.94	73.45	44.91	1.63
2002	950.29	1236.21	696.25	1.77	293.14	370.97	226.47	1.63	100.14	137.56	67.75	2.03	56	71.18	43.28	1.64
2003	956.18	1247.73	697.18	1.78	289.91	368.81	222.43	1.65	99.28	136.91	66.83	2.04	53.43	68.18	41.09	1.65
2004	971.91	1268.24	708.56	1.78	290.18	369.09	222.82	1.65	99.32	137.14	66.73	2.05	51.44	65.23	39.96	1.63
2005	929.7	1214.74	676.4	1.79	274.89	350.44	210.59	1.66	93.69	129.6	62.85	2.06	47.22	59.84	36.74	1.62
2006	882.62	1080.01	685.65	1.57	277.62	354.03	212.69	1.66	95.37	132.41	63.65	2.08	51.01	64.56	39.75	1.62
2007	826.14	1071.8	605.89	1.76	271.32	347.83	206.5	1.68	93.82	130.88	62.14	2.1	48.42	61.68	37.44	1.64
2008	830.36	1073.61	611.06	1.75	269.73	345.68	205.51	1.68	93.93	131.1	62.24	2.1	47.28	60.8	36.09	1.68
2009	811.47	1035.07	604.95	1.71	259.86	335.4	196.15	1.7	90.51	127.3	59.22	2.14	44.57	57.61	33.87	1.7
2010	813.4	1066.34	590.27	1.8	255.18	331.04	191.35	1.73	90.94	127.99	59.5	2.15	42.61	55.4	32.1	1.72
2011	826.19	1081.02	601.41	1.79	252	326.22	189.68	1.71	90.62	127.55	59.36	2.14	41.18	53.75	30.88	1.74
2012	823.73	1077.03	600.31	1.79	241.2	313.31	180.68	1.73	87.63	123.65	57.16	2.16	38.42	49.81	29.1	1.71
2013	795.49	1038.66	580.19	1.79	236.63	309.23	175.93	1.75	86.26	122.73	55.51	2.21	36.23	47.48	27.05	1.75
2014	774.08	1011.47	563.68	1.79	228.78	300.69	168.72	1.78	83.97	119.95	53.7	2.23	33.88	44.67	25.08	1.78
2015	769.28	998.84	565.55	1.76	225.53	295.14	167.47	1.76	83.8	119.39	53.89	2.21	33.17	43.45	24.83	1.74
2016	775.63	1009.32	568.33	1.77	227.99	300.67	167.64	1.79	84.5	120.76	54.15	2.23	32.82	43.31	24.32	1.78
2017	744.67	964.96	548.76	1.75	217.06	286.06	159.79	1.79	80.98	115.86	51.79	2.23	30.58	40.54	22.52	1.8
2018	730.66	946.19	538.78	1.75	210.38	278.82	153.66	1.81	78.88	113.24	50.18	2.25	30.23	40.31	22.09	1.82
2019	721.79	930.72	535.64	1.73	205.38	273.56	149.02	1.83	77.26	111.72	48.53	2.3	29.65	39.65	21.61	1.83
%	−30 *	−30 *	−30 *	0	−41 *	−36 *	−45 *	14 *	−31 *	−27 *	−37 *	13 *	−63 *	−60 *	−67 *	17 *

* *p* < 0.01; % means the percentage changes in DRaj in 2019 vs. DRaj in 1996; M/W: ratio men/women.

**Table 3 ijerph-19-12827-t003:** The crude death rate, the percentage change between 1996 and 2019, mean annual percentage change (AAPC), and 99% confidence limits (99%CL) for all causes of death and cardiovascular diseases per 100,000 population.

	All Causes of Death						Cardiovascular Diseases					
Age Group	1996	2019	% Change	AAPC (%)	99%CL		1996	2019	% Change	AAPC (%)	99%CL	
Total population												
35–39	300.42	197.93	−34	−1.8 *	−2.1	−1.5	45.03	25.92	−47	−2.4 *	−2.6	−2.1
40–44	396.28	256.92	−35	−1.9 *	−2.1	−1.7	84.67	46.59	−47	−2.7 *	−3.5	−1.8
45–49	540.39	366.81	−32	−1.6 *	−2.0	−1.3	145.27	82.49	−46	−2.4 *	−2.6	−2.2
50–54	770.39	549.14	−29	−1.5 *	−1.8	−1.1	238.53	140.98	−43	−2.2 *	−2.8	−1.5
55–59	1104.86	789.71	−29	−1.5 *	−1.8	−1.2	376.47	216.72	−44	−2.4 *	−2.7	−2.2
60–64	1708.75	1247.23	−27	−1.3 *	−1.5	−1.1	623.51	371.07	−41	−2.1 *	−2.2	−2.0
65–69	2564.01	1885.53	−26	−1.4 *	−1.6	−1.3	961.78	584.24	−39	−2.2 *	−2.6	−1.8
70–74	3990.69	2948.01	−26	−1.3 *	−1.7	−0.9	1548.07	942.15	−40	−2.0 *	−2.4	−1.7
Men												
35–39	436.05	274.91	−37	−1.9 *	−2.3	−1.5	53.95	31.93	−41	−2.3 *	−2.7	−1.9
40–44	551.57	345.64	−37	−2.0 *	−2.2	−1.8	104.83	58.50	−44	−2.6 *	−3.4	−1.7
45–49	719.94	489.43	−32	−1.6 *	−2.0	−1.3	179.69	104.09	−42	−2.4 *	−2.7	−2.1
50–54	1002.23	736.42	−27	−1.3 *	−1.7	−0.9	298.70	189.84	−36	−1.9 *	−2.5	−1.4
55–59	1411.88	1052.21	−25	−1.3 *	−1.6	−1.0	475.70	295.14	−38	−2.1 *	−2.4	−1.8
60–64	2173.83	1651.83	−24	−1.1 *	−1.3	−0.9	788.36	506.34	−36	−1.8 *	−1.9	−1.6
65–69	3240.26	2470.58	−24	−1.3 *	−1.4	−1.1	1195.81	779.52	−35	−1.8 *	−1.9	−1.7
70–74	5060.29	3858.00	−24	−1.2 *	−1.6	−0.7	1907.36	1249.21	−35	−1.7 *	−1.9	−1.5
Women												
35–39	171.13	122.89	−28	−1.4 *	−1.6	−1.2	36.53	20.06	−45	−2.5 *	−2.9	−2.2
40–44	248.24	173.18	−30	−1.7 *	−2.3	−1.1	65.45	35.35	−46	−2.6 *	−3.7	−1.5
45–49	369.21	254.79	−31	−1.5 *	−1.7	−1.3	112.45	62.76	−44	−2.5 *	−2.6	−2.4
50–54	551.42	381.76	−31	−1.6 *	−2.0	−1.2	181.71	97.33	−46	−2.5 *	−3.4	−1.5
55–59	821.84	561.60	−32	−1.7 *	−2.0	−1.4	285.00	148.58	−48	−2.8 *	−3.1	−2.5
60–64	1295.04	907.28	−30	−1.4 *	−1.6	−1.3	476.87	257.42	−46	−2.5 *	−2.6	−2.4
65–69	1990.76	1415.45	−29	−1.6 *	−1.7	−1.4	763.39	427.35	−44	−2.5 *	−2.8	−2.3
70–74	3141.90	2266.47	−28	−1.3 *	−1.9	−0.8	1262.95	712.18	−44	−2.3 *	−2.7	−1.9

* *p* < 0.01; % change means the percentage changes in DRaj in 2019 vs. DRaj in 1996.

**Table 4 ijerph-19-12827-t004:** The crude death rate, the percentage change between 1996 and 2019, mean annual percentage change (AAPC), and 99% confidence limits (99%CL) due to ischemic heart diseases and cerebrovascular diseases per 100,000 population.

	Ischemic Heart Disease						Stroke					
Age Group	1996	2019	% Change	AAPC	CI 99%		1996	2019	% Change	AAPC	CI 99%	
				(%)						(%)		
Total Population												
35–39	12.58	8.08	−36	−1.7 *	−2.4	−1.1	6.29	1.96	−69	−5.0 *	−5.6	−4.4
40–44	25.4	16.42	−35	−2.1 *	−2.9	−1.3	13.83	3.58	−74	−5.5 *	−5.9	−5
45–49	47	30.49	−35	−1.9 *	−2.4	−1.4	27.92	8.11	−71	−5.4 *	−5.8	−5.1
50–54	79.16	56.13	−29	−1.6 *	−1.7	−1.4	49.95	15.14	−70	−5.3 *	−5.7	−5
55–59	128.75	88.71	−31	−1.7 *	−1.9	−1.4	83.49	25.45	−70	−5.0 *	−5.3	−4.8
60–64	214.58	149.71	−30	−1.4 *	−1.5	−1.2	142.88	50.89	−64	−4.3 *	−4.6	−4.1
65–69	316.39	219.49	−31	−1.5 *	−1.6	−1.4	237.74	95.28	−60	−3.9 *	−4.1	−3.6
70–74	474.08	325.67	−31	−1.6 *	−1.8	−1.3	413.86	176.15	−57	−3.4 *	−3.6	−3.1
Men												
35–39	17.88	11.9	−33	−1.8 *	−2.4	−1.1	6.72	1.97	−71	−5.1 *	−6.2	−4
40–44	37.26	23.8	−36	−2.1 *	−2.9	−1.4	15.84	3.93	−75	−5.7 *	−6.4	−4.9
45–49	66.99	44.11	−34	−1.7 *	−2.1	−1.3	31.64	9.39	−70	−5.1 *	−5.7	−4.6
50–54	114.01	84.76	−26	−1.4 *	−1.6	−1.2	58.44	18.8	−68	−4.9 *	−5.9	−3.8
55–59	178.91	130.63	−27	−1.5 *	−1.8	−1.2	103.29	34.39	−67	−5.0 *	−6	−4
60–64	292.15	220.01	−25	−1.0 *	−1.2	−0.9	178.53	69.16	−61	−4.0 *	−5	−3
65–69	419.01	311.17	−26	−1.2 *	−1.3	−1.1	297.89	130.9	−56	−3.6 *	−4.6	−2.7
70–74	619	458.82	−26	−1.2 *	−1.5	−0.9	515.64	240.4	−53	−3.4 *	−3.9	−3
Women												
35–39	7.53	4.35	−42	−1.4 *	−1.8	−1	5.88	1.95	−67	−5.6 *	−6.5	−4.7
40–44	14.09	9.46	−33	−2.0 *	−3.1	−0.9	11.91	3.25	−73	−5.3 *	−7.1	−3.5
45–49	27.94	18.05	−35	−1.6 *	−1.9	−1.4	24.38	6.94	−72	−5.8 *	−6.7	−5
50–54	46.25	30.54	−34	−1.8 *	−2.2	−1.3	41.93	11.88	−72	−5.3 *	−6.4	−4.2
55–59	82.52	52.28	−37	−1.8 *	−2	−1.6	65.23	17.68	−73	−5.5 *	−6.4	−4.7
60–64	145.58	90.64	−38	−1.9 *	−2	−1.7	111.17	35.54	−68	−4.7 *	−5.7	−3.7
65–69	229.4	145.83	−36	−1.9 *	−2	−1.8	186.74	66.66	−64	−4.6 *	−5.7	−3.5
70–74	359.08	225.95	−37	−1.9 *	−2	−1.8	333.09	128.04	−62	−4.1 *	−4.7	−3.5

* *p* < 0.01; % change means the percentage changes in DRaj in 2019 vs. DRaj in 1996.

## Data Availability

The datasets used and/or analyzed during the current study are available from the corresponding author on reasonable request.

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
