# Peer review of "Sex Differences in Cardiovascular Disease Mortality in Brazil between 1996 and 2019"

_ijerph, 2022, doi:10.3390/ijerph191912827_

Round 1

Reviewer 1 Report

·       The title is not appropriate as used here. Suggest- “Sex differences in cardiovascular diseases mortality in Brazil between 1996 and 2019”.

·         Line 24: “and stroke in men and women aged……

·    Line 35-36: Please provide a strong conclusions/implication of this study and directions for future studies.

·         Please add “men” to the keywords list.

·         The introduction is short and very weak in its current form. Few studies are used to support the aim of the study. I do believe many studies are conducted on this topic. This section could be strengthen by including the following (1) the major risk factors for CVD in Brazil and worldwide; (2) the reasons women have lower rates of CVD than men; and (3) the importance/significance of this work. What is new? Why this study is important in light of previous studies? This should be clearly stated at the end of the section.

·         Why P-values are not included in Table 2?

·         Table 4 should be better organized same as Table 3.

·     The discussion is also very short and devoted of literature. Even though authors include some discussion, there are definitely results that are not discussed. The authors should check all their interpretations of the results.

·         Line 255-263: Limitations of the study should be discussed in depth. Please focus on the limitations related to the statistical analysis used.     

·      English language used could be improved. The paper should be edited by an English language speaker.    

Reviewer 2 Report

The same authors have already published  many papers on the same issue changing a little bit the period of enrollment, one in the last year, and one in the last 5 years:
- Mansur AP, Favarato D. Cardiovascular and Cancer Death Rates in the Brazilian Population Aged 35 to 74 Years, 1996-2017. Arq Bras Cardiol. 2021 Aug;117(2):329-340. English, Portuguese. doi: 10.36660/abc.20200233. PMID: 34495229; PMCID: PMC8395784.
- Mansur Ade P, Favarato D. Trends in Mortality Rate from Cardiovascular Disease in Brazil, 1980-2012. Arq Bras Cardiol. 2016 Jul;107(1):20-5. doi: 10.5935/abc.20160077. Epub 2016 May 24. PMID: 27223642; PMCID: PMC4976952.
Moreover other 5 paper on the same issue between 2010 and 2002.
- Mansur Ade P, Favarato D, Avakian SD, Ramires JA. Trends in ischemic heart disease and stroke death ratios in brazilian women and men. Clinics (Sao Paulo). 2010;65(11):1143-7. doi: 10.1590/s1807-59322010001100016. PMID: 21243288; PMCID: PMC2999711.
- Mansur Ade P, Lopes AI, Favarato D, Avakian SD, César LA, Ramires JA. Epidemiologic transition in mortality rate from circulatory diseases in Brazil. Arq Bras Cardiol. 2009 Nov;93(5):506-10. doi: 10.1590/s0066-782x2009001100011. PMID: 20084313.
- Mansur Ade P, de Souza Mde F, Timerman A, Avakian SD, Aldrighi JM, Ramires JA. Trends in the risk of death from cardiovascular, cerebrovascular and ischemic diseases in thirteen States of Brazil from 1980 to 1998. Arq Bras Cardiol. 2006 Nov;87(5):641-8. English, Portuguese. doi: 10.1590/s0066-782x2006001800015. PMID: 17221042.
- de Souza Mde F, Alencar AP, Malta DC, Moura L, Mansur Ade P. Serial temporal analysis of ischemic heart disease and stroke death risk in five regions of Brazil from 1981 to 2001. Arq Bras Cardiol. 2006 Dec;87(6):735-40. English, Portuguese. doi: 10.1590/s0066-782x2006001900009. PMID: 17262111.
- Mansur Ade P, Souza Mde F, Timermann A, Ramires JA. Trends of the risk of death due to circulatory, cerebrovascular, and ischemic heart diseases in 11 Brazilian capitals from 1980 to 1998. Arq Bras Cardiol. 2002 Sep;79(3):269-84. English, Portuguese. doi: 10.1590/s0066-782x2002001200007. PMID: 12386729. Conclusion are always the same no novelty could be add to this isssue.

Reviewer 3 Report

The authors analyzed the trends of all-cause mortality, CVD, ischemic heart disease, and stroke in men and women in Brazil from 1996 to 2019. I have the following comments:

Section Abstract, conclusions line 35,36: Needs reformulation, the ratio men to women for CVD is repeated in the sentence

Introduction is too short and poorly referenced. L58-61, Aim of the study – what is the novelty of your research?

L75. Number as 2.1.  Statistical analysis subsection

Section 2. Ethical requirements? They are missing

Section Results

L200. Please explain: In women, the reduction was more significant for IHD than for stroke (p<0.001). 

According to table 4 in women; crude death rate, the percentage change and annual percentage change (AAPC) seems to be more important in women for stroke than for IHD?

Moreover, section Results it is not particularly well written, it is difficult for the reader to follow the results. 

Tables must be set according to Instructions for authors. 

Under each table/figure, ALL abbreviations used in that table/figure must be explained – see the Instructions for authors regarding abbreviations.

Figures are blurred and some of them unreadable. Adjust the size of them

Many empty lines through the manuscript – must be removed.

L142. No point before (Table 2)

Table 3. No empty cells are allowed in a scientific paper. Set the table properly as to avoid the empty cells.

Table 4 must be in word, no photo. Same comments as above.

Section Discussion

Line 211. The reference number 7 should be in square brackets

Some of the sentences need to be reformulated: for example, L231: The proportion of CVD, IHD, and stroke was consistently higher in men than in women, and this proportion was higher for IHD; L 238: possible hypotheses to justify the increasing increase in IHD and stroke in men. 

Since the burden of diabetes is expected to grow exponentially in the next years (https://doi.org/10.3390/ijerph17124456), diabetes will be an increasingly important factor for ischemic heart disease related mortality. A more detailed description about the probable influence of the increasing burden of diabetes on CVD and ischemic heart disease mortality, since now Brazil’s overall ranking is around fifth among the top countries with diabetes; I suggest checking for information and referring to the following published papers https://doi.org/10.3390/diagnostics10050314

https://doi.org/10.1016/j.biopha.2022.112772

https://www.spandidos-publications.com/10.3892/etm.2020.8714

L266. Remove „In conclusion,” as it is obvious and repetitive; Justify the entire paragraph of Conclusions section. 

References must be in the MDPI style – check the Instructions for authors.

Extensive English revision and editing of the text.

Round 2

Reviewer 1 Report

No further comments.

Author Response

Thanks again for your time in reviewing my manuscript. We also edit the minor corrections in the English language.

Reviewer 2 Report

A novel paper on CVD death in the same country could be useful if the last was far at least 5 years. 

Author Response

Thanks again for your time in reviewing my manuscript. I'm afraid I disagree with the reviewer's statement, "A novel paper on CVD death in the same country could be useful if the last was far at least 5 years". As I exampled previously, there is a publication of the US CVD death rate in one important cardiologic Journal every year or two years. So why not do the same with our Brazilian data every two or three years, monitoring and updating the most important epidemiological changes?

Reviewer 3 Report

The authors responded to all my request that "OK. Done" or similar. However, the corrections to the manuscript remained poor and insufficient. Please check my previous report and pro ceed consequently. Respond to each of my request and also check the Instructions for authors regarding formatting tables, aspect of the figures, information for references and the way they are provided.

Author Response

Thanks again for your time in reviewing my manuscript. We edited the English language as suggested. We think we objectively referenced the manuscript; new references could be repetitive and not add further information.